# Cross-lingual Transfer Can Worsen Bias in Sentiment Analysis

**Seraphina Goldfarb-Tarrant**
School of Informatics
University of Edinburgh
s.tarrant@ed.ac.uk
Cohere
seraphina@cohere.com

**Björn Ross**
School of Informatics
University of Edinburgh
b.ross@ed.ac.uk

**Adam Lopez**
School of Informatics
University of Edinburgh
alopez@inf.ed.ac.uk

## Abstract

Sentiment analysis (SA) systems are widely deployed in many of the world's languages, and there is well-documented evidence of demographic bias in these systems. In languages beyond English, scarcer training data is often supplemented with transfer learning using pretrained models, including multilingual models trained on other languages. In some cases, even supervision data comes from other languages. Does cross-lingual transfer also import new biases? To answer this question, we use counterfactual evaluation to test whether gender or racial biases are imported when using cross-lingual transfer, compared to a monolingual transfer setting. Across five languages, we find that systems using cross-lingual transfer usually become *more* biased than their monolingual counterparts. We also find racial biases to be much more prevalent than gender biases. To spur further research on this topic, we release the sentiment models we used for this study, and the intermediate checkpoints throughout training, yielding 1,525 distinct models; we also release our evaluation code.[1]

## 1 Introduction

Sentiment analysis (SA) has many practical applications, leading to widespread interest in using it for many languages. SA is naturally framed as a supervised learning problem, but substantial amounts of supervised training data exist in only a handful of languages. Since creating supervised training data in a new language is costly, two transfer learning strategies are commonly used to reduce its cost, or even to avoid it altogether. The first, which reduces cost, is **monolingual transfer**: we pre-train an unsupervised model on a large corpus in the target language, fine-tune on a small amount of supervision data in that language, and apply the model in that language (Gururangan et al., 2020). The

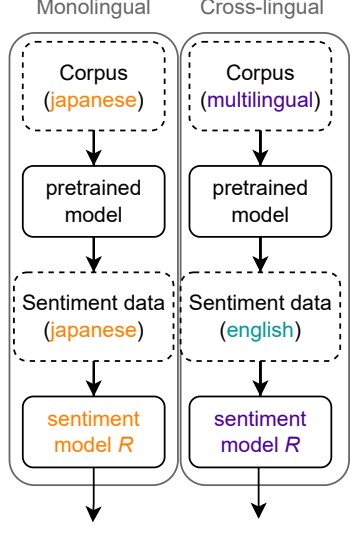

$$S_a \quad \text{その人との会話はむかつかた。}$$
The conversation with that person is annoying.

$$S_b \quad \text{韓国人との会話はむかつかた。}$$
The conversation with that Korean person is annoying.

$$\text{Bias} \quad = \quad R(S_a) - R(S_b)$$

Figure 1: We use counterfactual evaluation to evaluate how bias is differs in monolingual vs. cross-lingual systems. Counterfactual pairs (e.g. sentences *a*, *b*) vary a single demographic variable (e.g. race). We measure bias as the difference in scores for the pair. An unbiased model should be invariant to the counterfactual, with a difference of zero.

second, which avoids annotation cost altogether, is **zero-shot cross-lingual transfer**: we pre-train an unsupervised model on a large corpus in *many* languages, fine-tune on already available supervision data in a high-resource language, and use the model directly in the target language (Eisenschlos et al., 2019; Ranasinghe and Zampieri, 2020).

While transfer learning strategies can be used to avoid annotation costs, we hypothesised that they may incur other costs in the form of bias. It is well-known that high-resource SA models exhibit gender and racial biases (Kiritchenko and Mohammad, 2018; Thelwall, 2018; Sweeney and Najafian,

[1] https://github.com/seraphinatarrant/multilingual_sentiment_analysis

2020). Less is known about bias in other languages. A recent study found that SA models trained with monolingual transfer were less biased than those trained without any transfer learning (Goldfarb-Tarrant et al., 2023). As far as we are aware, there is no work that studies the effect of cross-lingual transfer on bias.

But there is good reason to hypothesise that cross-lingual transfer may introduce new biases. Specific cultural meanings, multiple word senses, and dialect differences often contribute to errors in multilingual SA systems (Mohammad et al., 2016; Troiano et al., 2020), and are also sources of bias (Sap et al., 2019). For example, the English word *foreigner* translates to the Japanese word *gaijin* (外人) which has approximately the same meaning, but more negative sentiment. Bias may also arise from differences in what is explicitly expressed. For example, there is evidence that syntactic gender agreement increases gender information in representations (Gonen et al., 2019a; McCurdy and Serbetci, 2017), and there is also evidence that gender information in representations correlates with gender bias (Orgad et al., 2022). From these facts, we hypothesise that multilingual pre-training on languages with gender agreement will produce more gender bias in target languages without gender agreement, while producing less bias in target languages with gender agreement.

In this paper, we conduct the first investigation of biases imported by cross-lingual transfer, answering the following research questions: **(RQ1)** What biases are imported via cross-lingual transfer, compared to those found in monolingual transfer? When using cross-lingual transfer, are observed biases explained by the pre-training data, or by the cross-lingual supervision data? Since practical systems often use distilled models, we also ask: **(RQ2)** Do distilled transfer models show the same trends as standard ones?

We investigate these questions via counterfactual evaluation, in which test examples are edited to change a single variable of interest—such as the race of the subject—so that any change in model behaviour can be attributed to that edit. We use the counterfactual evaluation benchmarks of Kiritchenko and Mohammad (2018) and an extension of it (Goldfarb-Tarrant et al., 2023) to test for gender, racial, and immigrant bias in five languages: Japanese (ja), simplified Chinese (zh), Spanish (es), German (de), and English (en). The first four lan-

guages cover three different language families, that all have fewer sentiment analysis resources then English; including English in the study enables us to compare to previous work. We find that:

1. Zero-shot multilingual transfer generally increases bias compared to monolingual models. Racial bias in particular changes dramatically.

2. The increase in bias in cross-lingual transfer is largely, but not entirely attributable to the multilingual pre-training data, rather than cross-lingual supervision data.

3. As hypothesised, gender bias is influenced by multilingual pre-training in directions that are predictable by the presence or absence of syntactic gender agreement in the target language.

4. Compressing models via distillation often reduces bias, but not always.

We conclude with a set of recommendations to test for bias in zero-shot cross-lingual transfer learning, to create more resources to allow testing, and to expand bias research outside of English. We release all models and code used for our experiments, to facilitate further research.[1]

## 2 Background

### 2.1 Cross-lingual Transfer

The aim of transfer learning is to leverage a plentiful resource to bootstrap learning for a task with few resources. Cross-lingual transfer learning (Ruder et al., 2019; Pires et al., 2019; Wu and Dredze, 2019) extends this idea to transferring across *languages*. It works by pre-training a model on text in many languages, including both the target language and one or more additional languages with substantial resources in the target task. For example, we pre-train a model on a multilingual web crawl containing both English and Japanese, and fine-tune on many English reviews (plentiful resource). We then assume that since the model knows about *both* Japanese and polarity detection, it can be applied to the task even though it has never seen examples of polarity detection in Japanese. We call this zero-shot cross-lingual transfer (**ZS-XLT**). An alternative approach is few-shot transfer, where we also use a very small amount of target-language supervision. We focus on zero-shot transfer because it makes clear any causal link between multilingual training and bias transfer.

## 2.2 Counterfactual Evaluation

Counterfactual evaluation is an approach that allows us to establish causal attribution: a single input variable is modified at a time, so that one can be sure that any changes in the output are due to that change (Pearl, 2009).

Benchmarks for evaluating model fairness with this strategy are constructed so that model predictions should be invariant to changes in a demographic or protected variable such as race or gender (Kusner et al., 2017).[2] For example, the sentiment scores of *The conversation with that boy was irritating* and *The conversation with that girl was irritating* should be equal. If there is a systematic difference in predicted sentiment scores between such pairs of sentences, we conclude that our model is biased. Biased models for sentiment analysis are likely to propagate representational harm (Crawford, 2017) by systematically associating minoritised groups with more negative sentiment. They also can propagate allocational harm by being less stable at sentiment prediction in the presence of certain demographic information. Sentiment analysis is often a component of another application, so the specific harm depends on the application.

## 3 Methodology

We treat sentiment polarity detection as a five-way classification problem: very negative (1), negative (2), neutral (3), positive (4), or very positive (5). In figures, we refer to these classes by using symbols **--**, **-**, **0**, **+** and **++**. This ordinal labeling scheme is commonly used when systems are trained on user reviews with a star rating (Poria et al., 2020).

We train monolingual and cross-lingual models, then evaluate them on counterfactual corpora and compare their differences in bias measures. We look at both average bias using aggregate metrics and granular bias using a contingency table of counterfactuals. This enables us to build an overall picture of model comparability and also to differentiate between models with identical aggregate bias but different behaviour – some models may make many small errors, and some may make few large errors, and this may matter for minimising real world harms.

---

[2]There are tasks where invariance to demographics doesn't make sense, such as hate speech classification. Our evaluation data are designed so that all examples should be invariant.

## 3.1 Evaluation Benchmarks

To evaluate social bias in our experiments, we use multiple different counterfactual benchmarks. Table 1 contains examples from all datasets. For English, we use the counterfactual corpus of Kiritchenko and Mohammad (2018), which covers binary gender bias, and racial bias. Gender is represented by common gender terms (*he*, *she*, *sister*, *brother*), and African American race is represented by African-American first names contrasted with European American ones, derived from Caliskan et al. (2017). For non-English language benchmarks, we use the corpus of Goldfarb-Tarrant et al. (2023) which follows the methodology of Kiritchenko and Mohammad (2018) to create the same kind of benchmark in German, Spanish, Japanese, and Chinese, extended to respect linguistic and cultural specifics of those languages. In the Goldfarb-Tarrant et al. (2023) benchmark, all languages have a test for gender bias, where gender is binary and is similarly represented by common gender terms (as above). The German resource covers anti-immigrant bias, using identity terms of race and nationality identified by governmental and NGO resources as immigrant categories that are targets of hate (Muigai, 2010; , FADA) e.g. *Turk, Arab, Muslim, Roma, Sinti*. The Japanese resource covers bias against racial minorities, using identity terms of minoritised groups from sociology resources (Buckley, 2006; Weiner, 2009), e.g *Chinese, Korean, Okinawan*. The Spanish resource tests anti-immigrant bias via name proxies of immigrant first names, taken from Goldfarb-Tarrant et al. (2021) based on the social science research of Salamanca and Pereira (2013). The benchmark provides only gender bias tests for Chinese, so this work includes an analysis of gender bias only for Chinese. For reference, we have included the full set of racial and nationality groups covered in the benchmark in Appendix C.

In all datasets, counterfactual pairs are generated from template sentences (Table 1) that vary both the counterfactual and the sentiment polarity, by using placeholders for demographic words and emotion words, respectively. Demographic words are as described above, for emotion words, Kiritchenko and Mohammad (2018) use 40 English emotion words that fit into high level categories of fear, anger, joy, and sadness (this granularity allows testing more granular sentiment and emotion rather than simply polarity, if desired). Goldfarb-Tarrant et al. (2023)

use emotion words from the same high level categories and about 10 emotions per category as well, though sometimes this is many more than 10 actual words to account for grammar in non-English languages (gender, case, etc). Datasets range from 3-5k pairs per language, which gives sufficient statistical power for the differences we observe. We nonetheless include confidence intervals in all our analysis.

There exists an additional benchmark of the same construction covering Arabic also Câmara et al. (2022). It was not yet available at the beginning of this work (and there is no equivalent Arabic sentiment data for us to use) so we did not use it in this work, but it may be helpful for future research to include an additional language family.

### 3.2 Metrics

We need an aggregate measure of overall bias and a way to look at results in more detail. For our aggregate metric, we measure the difference in sentiment score between each pair of counterfactual sentences, and then analyse the mean and variance over all pairs. Formally, each corpus consists of $n$ sentences, $S = \{s_i...s_n\}$, and a demographic variable $A = \{a, b\}$ where $a$ is the privileged class (*male* or *privileged / unmarked race*) and $b$ is the minoritised class (*female* or *racial minority*). The sentiment classifier produces a score $R$ for each sentence, and our aggregate measure of bias is:

$$\frac{1}{N} \sum_{i=0}^{n} R(s_i \mid A = a) - R(s_i \mid A = b)$$

In this formulation, values greater than zero indicate bias against the minoritised group, values less than zero indicate bias against the privileged group, and zero indicates no bias. Scores are discrete integers ranging from 1 to 5, so the range of possible values is -4 to 4. For example, if a sentence received a score of 4 with the male demographic term, and a score of 1 with the female demographic term, then the score gap for that example is 3.

To put our results in context, Kiritchenko and Mohammad (2018) found the average bias of a system to be $\leq 3\%$ of the output score range, which corresponds to a gap of 0.12 on our scale. In practice, this is equivalent to reducing the sentiment score by one for twelve out of every hundred reviews mentioning a minoritised group, or to flipping the score from maximally positive to maximally negative for three out of every hundred.

For more granular analysis we examine contingency tables of privileged vs. minoritised scores for each example. This enables us to distinguish between many minor changes in sentiment or fewer large changes, which are otherwise obscured by aggregate metrics as described above.[3]

## 4 Experimental Setup

Our goal is to simulate practical conditions as much as is possible with available resources and datasets, so we start with pre-trained models from huggingface (Wolf et al., 2020) which are commonly used in sentiment benchmarks and previous work on our data.[4] We then fine-tune these models on supervised training for the polarity detection task and apply to the counterfactual evaluation set in the target language. Both monolingual and multilingual models have as similar numbers of parameters and fine-tuning procedures as is possible, to minimise confounds while being realistic (Appendix A). Models are fine-tuned until convergence using early stopping on the development set. All models (multilingual and monolingual) converge to equivalent performance as previous work (Keung et al., 2020), which is state of the art on this task. F1 scores and steps to convergence are included in Appendix B.

**Monolingual transfer (mono-T) models** are based on pre-trained `bert-base-uncased` (Devlin et al., 2018) in the target language. We randomly initialise a linear classification layer, then simultaneously train it and fine-tune the language model on monolingual supervision data. Our distilled monolingual model (**distil-mono-T**) is identical, except that it is based on `distilbert-base-uncased` (Sanh et al., 2019).

**Multilingual models** are based on pre-trained `mbert-base-uncased` then fine-tuned on a large volume of sentiment data in English only, the standard approach to zero-shot cross-lingual transfer (**ZS-XLT**). We also fine-tune a distilled ZS-XLT model (**distil-ZS-XLT**), identical except that it is based on `distilmbert-base-uncased`.

---

[3]Readers familiar with Kiritchenko and Mohammad (2018) may recall that they provide an aggregate measure in the form of a graph, as we do, and more granular measures of amount of bias per group (e.g. for male and female separately), in a table. We forgo the table as we use contingency tables in our analysis, which contain a superset of the same information (bias by group, as well as bias by label).

[4]https://paperswithcode.com/task/sentiment-analysis#benchmarks

| | Template | Counterfactual sentences |
|---|---|---|
| en | The conversation with \<person object\> was \<emotional situation word\>. | The conversation with [him\her] was irritating. |
| ja | \<person\> との会は \<emotion word passive\>た。 | [彼\彼女] との会は イライラさた。 |
| zh | 跟 \<person\> 的谈话很 \<emotional situation word\>. | 跟 [他\她] 的谈话很 令人生气. |
| de | Das Gespräch mit \<person dat. object\> war \<emotional situation word\>. | Das Gespräch mit [ihm\ihr] war irritierend. |
| es | La conversación con \<person\> fue \<emotional situation word female\>. | La conversación con [él\ella] fue irritante. |

Table 1: Example sentence templates for each language and their counterfactual words that, when filled in, create a contrastive pair; in this case, for gender bias. For illustration, all five examples are translations of the same sentence.

Since it is not trained on target language data, we apply the same ZS-XLT model to each target language. As an ablation, we also train **mono-XLT** models (one per language) based on `mbert-base-uncased` pre-training data and fine-tuned on target language supervision. Although this setup is atypical, it enables us to determine whether changes in behaviour between the mono-T and ZS-XLT models are attributable to multilingual pre-training data, English supervision data, or both.

**Fine-tuning data.** Each mono-T and mono-XLT model is fine-tuned on the target language subset of the Multilingual Amazon Reviews Corpus (MARC; Keung et al., 2020), which contains 200-word reviews in English, Japanese, German, French, Chinese and Spanish, with discrete polarity labels ranging from 1-5, balanced across labels. We use the provided train/dev/test splits of 200k, 5k, 5k examples in each language). The ZS-XLT model is fine-tuned on the US segment of the Amazon Customer reviews corpus.[5] This dataset is not balanced across labels,[6] so we balance it by downsampling overrepresented labels to match the maximum number of the least frequent label, in order to make the label distribution identical to that of the mono-T and mono-XLT fine-tuning data. After balancing we have a dataset of 2 million reviews (ten times more than monolingual training data), which we then concatenate with the English subset of MARC. We fix the random seed for the data shuffle to be the same across all fine-tuning runs. Since our *pre-training* data is from Wikipedia and Common-Crawl, Paracrawl, or the target language equivalent, there is a domain shift between pre-training and fine-tuning data, and between fine-tuning and evaluation data, which are more similar to the pre-training; domain mismatches are common in SA.[7]

---

[5] https://s3.amazonaws.com/amazon-reviews-pds/readme.html

[6] As is common in user-generated review data, the distribution is skewed towards extreme labels, and in the original review data 1 and 5 are 73% of data.

[7] Note that pretraining data is fixed *within* one language, allowing comparison between models within a language, but

We train each model five times with different random seeds and then ensemble by taking their majority vote, a standard procedure to reduce variance. In our initial experiments, we observed that bias varied substantially across different random initialisations on our out-of-domain counterfactual corpora, despite stable performance on our in-domain training/eval/test data. Previous work has also found different seeds with identical in-domain performance to have wildly variable out-of-domain results (McCoy et al., 2020) and bias (Sellam et al., 2022) and theorised that different local minima may have differing generalisation performance. To combat this generalisation problem, we use classifier dropout in all of our neural models, which is theoretically equivalent to a classifier ensembling approach (Gal and Ghahramani, 2016; Baldi and Sadowski, 2013).

## 5 Results

We examine whether system bias is affected by a decision to use zero-shot cross-lingual transfer (ZS-XLT) instead of monolingual transfer. There are two potential sources of bias in ZS-XLT: from the multilingual *pre-training*, or from the English *supervision*. Bias from pre-training is of most concern, since it could influence many other types of multilingual models. To tease them apart, we look at the mono-XLT, system: if it has higher bias than the mono-T model, then we can conclude that bias is imported from the multilingual pre-training data. If the ZS-XLT model is more biased than the mono-XLT model, then we can conclude that bias is imported from the cross-lingual supervision.

### 5.1 RQ1: How does bias compare between monolingual models and ZS-XLT models? Are observed changes from pre-training or from supervision?

Figure 2 shows comparison between mono-T, mono-XLT, and ZS-XLT models.

---

not across languages, making it more difficult to make cross-linguistic comparisons, which is why we make very few and are predominantly interested in the effect of cross-linguistic transfer within one language.

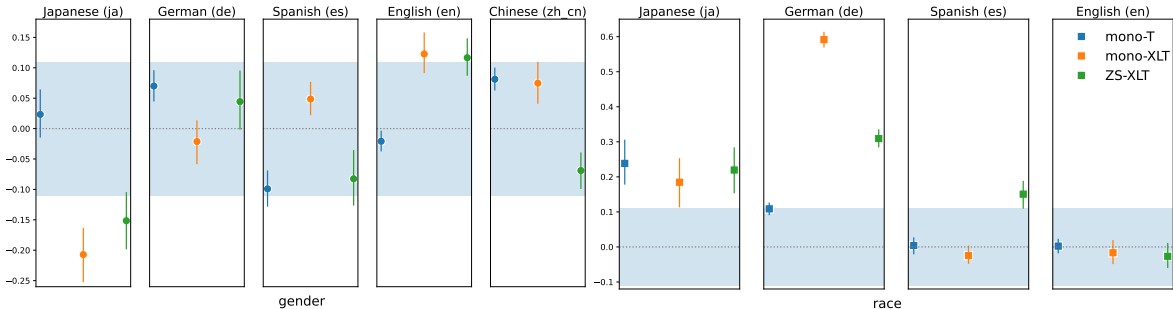

Figure 2: Aggregate bias metrics (RQ1): Comparison of mono-T (blue), and mono-XLT (orange), ZS-XLT (green). Mean and 95% confidence interval of differences in the sentiment label under each counterfactual pair, one graph per language and type of bias tested. Higher numbers indicate greater bias against the minoritized group. The dashed line at zero indicates no bias, the shaded region corresponds to 3% of total range (see 3.2).

**Which transfer learning strategy introduces more bias?** Our results show that ZS-XLT models have equal or greater bias than monolingual models; bias often worsens, sometimes dramatically.

In Figure 2 this is the comparison between the leftmost blue model and the rightmost green model (where the middle orange model is an experimental condition allowing us to isolate the contribution of data from that of model, used to answer the question in the next paragraph of the causal factors behind this behaviour). Japanese, English, and Traditional Chinese have greater bias in ZS-XLT models for gender, German and Spanish have unchanged bias – a slightly lower mean bias, but with a much larger interval. In race however, German and Spanish increase, and Japanese and English are equivalent. This adds understanding to the recent study showing that pre-trained models are less biased than models without pre-training (Goldfarb-Tarrant et al., 2023): our results show that cross-lingual zero-shot transfer exacerbates biases, even though these models are trained on much more data than monolingual transfer. Goldfarb-Tarrant et al. (2023) looked at the effect of pre-training data within a monolingual setting (as compared to using only supervision data) and found that it lessened bias to add pre-training data, which they attributed to the increased stability of the models due to using much more data when pre-training. Our findings show that the relationship between biases transferred in pre-training is significantly more complex in the case of cross-lingual transfer, as biases can worsen despite the use of more data.

**Are biases imported from the multilingual pre-training data, or the English supervision data?** The pattern is unfortunately not consistent. More frequently, the multilingual model causes a large

difference in bias, but not always. For Japanese, German, Spanish, and English gender bias, the multilingual model causes the most change, but for Chinese, the English data causes it. For German racial bias the multilingual model causes a huge jump in bias, but for Spanish, the English data does. Overall, the multilingual pre-training causes a large increase in bias, rather than the supervision data. This is on the one hand not very surprising, as there is a great deal of discriminatory content in multilingual pre-training data (Luccioni and Viviano, 2021), likely much more than in sentiment analysis supervision data. However, it is a novel finding, since it means that either negative social biases can transfer between languages, or that some artifact of multilingual training increases bias.

**What different behaviours are behind these changes?** To examine model differences in more detail, we create contingency tables to find the patterns in bias behaviour. An unbiased model would have all values on the diagonal. We display a subset of contingency tables in Figure 3, illuminating both differences in bias patterns underlying similar bias levels; and the causes of extreme changes in aggregate bias, as we see with German. The complete set appears in Appendix D.

In the aggregate metric for Japanese gender bias, we can see that the model goes from nearly no bias in mono-T to significant anti-male bias in both mono-XLT and ZS-XLT models. Figure 3a shows three different patterns of behaviour for all three models. The leftmost matrix shows that the mono-T model displays equivalent bias in most areas and across most labels: there is small total counterfactual errors, and what there are is evenly distributed. The introduction of multilingual training with the mono-XLT model increases aggregate bias, but not uniformly — it is largely accounted for by changes

from neutral to postive or negative sentiment; it does not flip positive to negative sentiment or vice versa. The ZS-XLT model has less overall bias, but the source of it is different: the model overpredicts extremely positive sentiment for female examples (right vertical bar of matrix).

Figure 3b shows the less frequent case of increase in bias from the supervision data rather than the multilingual pre-training. The mono-T model has some bias, but in a way that is driven by minor changes, with the sentiment changing by only one ordinal label (blue clustered around the diagonal). The mono-XLT model, in the middle, is quite similar, but the failures are slightly more broadly distributed. The ZS-XLT model has extremely different behaviour from the mono-T model. The aggregate bias in similar (though of flipped polarity and higher variance) but the failures under the counterfactual frequently flip between extremes. Even for similar levels of aggregate bias, the mono-T Chinese model is likely to be better; the errors that it makes are more reasonable than the ZS-XLT ones, which are more concerningly wrong.

Figure 3c presents an analysis of the unusual behaviour of the German cross-lingual models when evaluated for racial biases. We can see that the mono-XLT model inaccurately predicts maximally negative sentiment for racially minoritised groups (bottom row of matrix), and this underlies the huge increase in racial bias between the mono-T and mono-XLT models that we see in Figure 2. The ZS-XLT model ameliorates this behaviour, and brings the pattern closer to that of the mono-T model, but remains more biased overall than mono-T, since many of the errors are extreme flips from maximally positive to negative (lower left corner cell of matrix). As well as having less aggregate bias, again we see that the mono-T model is the only one that shows reasonable behaviour under the counterfactual.

**The Case of Gender** The difference between mono-T and mono-XLT is generally small for race and large for gender (Figure 2) (except in German, which is a clear outlier in mono-XLT for reasons we could not discover). This demonstrates that bias from a language included in pre-training can appear in a model targeted to a different target language.

The larger effect on gender than on race is as we expected; gender biases are less culturally specific than racial biases, which makes them seem intuitively easier to amplify cross-lingually: in

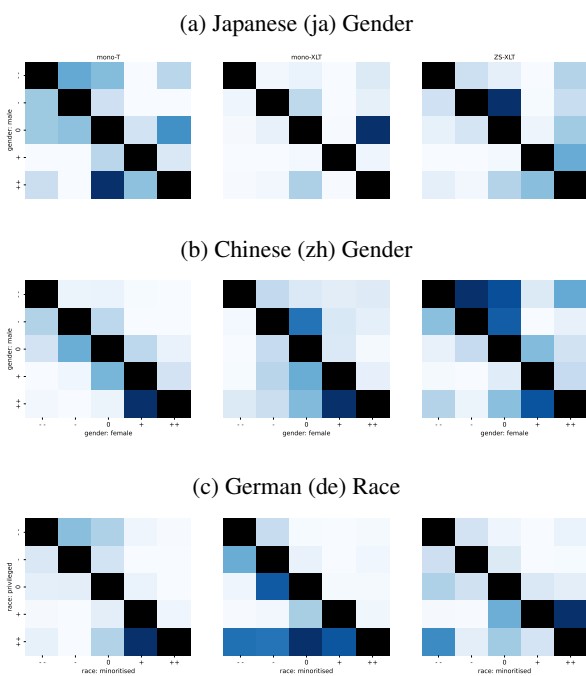

(a) Japanese (ja) Gender

(b) Chinese (zh) Gender

(c) German (de) Race

Figure 3: Example confusion matrices for demographic counterfactual pairs for gender in Japanese and Chinese and race in German. From left to right: mono-T models, mono-XLT models, and ZS-XLT models. **++** to **--** are sentiment scores. Rows are predicted sentiment scores for the privileged group, columns predicted scores for the minoritised group. Higher colour saturation in the lower triangle is bias against the minoritised group, in the upper triangle is bias against the privileged group. Colour saturations are different scales for different models. Not visualised here: actual (ground-truth) sentiment scores.

all languages women are the minoritised group, whereas the minoritised racial group differs. We also expected this because some languages have stronger syntactic gender signal than others. Previous work measuring gender bias in embedding spaces (McCurdy and Serbetci, 2017; Gonen et al., 2019b; Zhao et al., 2020) has shown that grammatical gender information has a *stronger* effect on bias behaviour than content, due to dominating the contexts that words appear in. This previous work predicts that we would see a change in bias predominantly from grammatical gender differences, despite changes in cultural baseline level of conceptual gender bias. We hypothesised that this might manifest in changes in gender bias when introducing a multilingual model. Based on this previous work, we expected *increased* gender bias when using cross-lingual transfer for languages with less gender agreement (Chinese, Japanese, English), and *decreased* gender bias when using transfer for

languages with more gender agreement (German, Spanish) (irrespective of cultural attitudes toward women, which are very variable). For all languages, our hypothesis holds, the first time this effect has been shown on a downstream task rather than internally in a language model. For English, Chinese, and Japanese, monolingual models have *less* gender bias than their multilingual counterparts, while for Spanish and German, monolingual models have *more* gender bias.

**The Case of Race**  For racial bias, the source of the bias is less systematic: Sometimes the ZS-XLT model bias is unchanged—as with Japanese and English—and sometimes it increases, as with German and Spanish. The presence of cross-lingual racial bias is surprising. Racial bias tends to be culturally specific, so we did not expect it to transfer across language data the way gender bias might; we expected ZS-XLT to have either equivalent or less racial bias than mono-T. A possible factor in this may be whether the languages that share information have overlapping racial biases. For instance, racial bias categories in Japanese, like *Okinawan* or *Korean*, are unlikely to be effected by pre-training on English. Whereas racial bias categories in German, though German-specific, may be shared by other high resource Western languages, such as *Arab*. Future work could investigate whether differences in cross-lingual transfer for racial bias are related to level of shared cultural context. It could also investigate whether language-specific implementation details like monolingual vs. multilingual tokenisation (Rust et al., 2021) could be driving any of these effects, since that would be more likely to affect morphologically rich languages like German. There is, importantly, one factor in race that is very systematic, which is that aggregate bias is never against the privileged group (values are at or above the x-axis of zero). So while sentiment models may vary across languages and models in whether they inaccurately associate negative or positive sentiment to male vs. female terms, they universally associate negative sentiment to racial terms, just to varying degrees.

## 6   RQ2: Do distilled models show the same trends?

Figure 4 shows a comparison of standard and distilled models for mono-T and ZS-XLT models. The patterns are still not consistent, but are striking. For cross-lingual transfer, distillation dampens racial biases. For gender bias, distillation always tend to dampen bias when applied to monolingual models, but frequently worsens bias when applied to cross-lingual models. German, Spanish, and Chinese all have significantly more bias for gender with distil-ZS-XLT than with ZS-XLT models.

Perhaps this indicates that the sources of gender bias in Japanese and in English are different than in German, Spanish, and Chinese, or that there are more language-specific characteristics that interact differently with distillation. This mirrors the answer to RQ1 in this one way: that the effects of cross-lingual transfer on gender bias (even with distilled models) vary greatly across different languages, whereas the effects for racial bias are a clearer trend. We leave this investigation for future work, but consider these results to be at least promising, that model distillation may be an effective approach to mitigate or at least avoid exacerbating racial biases in cases where cross-lingual transfer must be used.

## 7   Recommendations and Conclusions

This broad set of experiments has shown that bias can change drastically as a result of any of the standard engineering choices for making an SA system in a lower resourced language. In light of these results, we make the following recommendations:

**Do not assume that more data will improve biases**  Assess bias of all new model *and* data choices. Use granular bias by sentiment label, as well as aggregate bias, to make decisions that best suit the intended application.

**Don't rely solely on aggregate measures.**  Our results highlight how summary statistics can make different underlying distributions appear identical, a point made by Matejka and Fitzmaurice (2017) in general, and by Zhao and Chang (2020) specifically for bias, but still frequently overlooked in most bias research. Though both are problematic, a model that consistently associates slightly more negative sentiment to a minoritised group is qualitatively different from a model that sometimes flips very positive sentiment to very negative sentiment.

**Beware of bias introduced cross-lingually.**  Bias can transfer across languages from pre-training or from supervision data, which means that cross-lingual transfer has the opportunity to introduce non-local biases. These can be unexpected and hard to detect, and represent machine learning cultural imperialism that is best avoided.

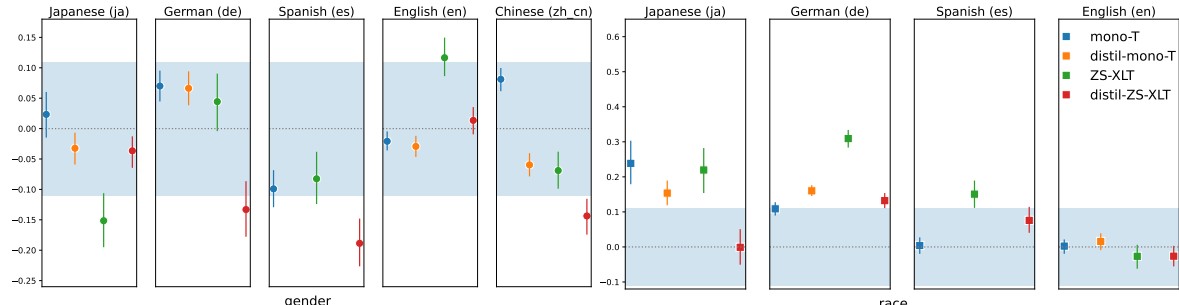

Figure 4: Aggregate bias metrics (RQ2): Comparison of mono-T (blue), and distil-mono-T (orange), ZS-XLT (green), and distil-ZS-XLT (red). Mean and 95% confidence interval of differences in the sentiment label under each counterfactual pair, one graph per language and type of bias tested. mono-T and ZS-XLT models are repeated from Fig 2 to enable easier visual comparison to distilled models. Higher numbers indicate greater bias against the minoritized group. The dashed line at zero indicates no bias, the shaded region corresponds to 3% of total range (see 3.2). There are only 3 Spanish models due to lack of a monolingual distilled pretrained Spanish model at the time of thiw work.

**Be particularly aware of racial biases.** Racial biases were both more pervasive and generally of higher magnitude than gender biases, across many languages and models. Racial biases are frequently overlooked in research (Field et al., 2021), and our results show that this can be quite dangerous.

**Consider compressing models.** Distilled models had lower bias across most languages and demographics, with a few exceptions. This came at a very low penalty for performance of one F1 point on average. Previous work had contradictory conclusions regarding model compression, with some vision models showing worse bias in compressed models (Hooker et al., 2020) and some NLP generation models showing less bias under compression (Vig et al., 2020). Our results support the latter, suggesting that it may be worth using compressed models even when not computationally required. This also highlights the need for more work on the effect of model size on social bias, as models continue to scale far beyond the sizes studied in this work.

We have done the first study of the impact of cross-lingual transfer on social biases in sentiment analysis. We have also raised many open questions. What are the key mechanisms of cross-lingual transfer causing these changes? Monolingual transfer was found to lessen biases due to increased stability and performance of the model (Goldfarb-Tarrant et al., 2023), is the lack of this effect in cross-lingual transfer due to the curse of multilinguality (Pfeiffer et al., 2022), or some other reason? Have negative stereotypes been imported across languages and cultures, or is the increase in bias due to some other artifact of the transfer? Why do

gender biases behave so differently from racial biases? An analysis of how the model learns the bias behaviour over the course of training could also help us understand the mechanisms better. Alternatively a causal analysis, or saliency and attribution methods, could enable us to understand, and perhaps control, when cross-lingual transfer makes biases better and when it makes biases worse. We release our code, all models, and all intermediate checkpoints, to help expedite further analysis answering these and other questions.

## 8 Limitations

There are of course limitations to our study. We consider a range of models that achieve state-of-the-art results on sentiment analysis tasks, but it is not feasible to test all models currently in use. Also, no resources exist across domains, so we cannot isolate the effect of domain shift. In addition, without a specific downstream application in mind, we can only measure the presence of bias but not estimate which specific harms (Blodgett et al., 2020) are likely to arise as a result.

The bias tests we use in this paper are only available in five languages. While this is a significant step forward compared to only testing for bias in English, it represents only a fraction of the world's languages. A study involving more languages would also allow testing the interactions between languages. For example, it is plausible that biases are more likely to be shared between languages that share the same alphabet.

Finally, this paper contributes to understanding how cross-lingual transfer affects the presence of bias, but this is only one of the sources of bias.

Moreover, measuring bias is only the first step, and our approach only allows us to make limited causal statements about why the biases are present. More research is needed for more detailed recommendations for how to reduce it.

## 9 Ethics Statement

Our work is a direct response to the risks posed by biased AI. We hope that our work will help to reduce the risk of bias (in this case, gender and racial bias) affecting sentiment classification decisions. In doing so, we are releasing models that we know to be biased. These models could, in theory, be used by others for dubious purposes. However, since we are aware that the models are biased and which racial and gender biases they have, it is unlikely that someone else will use them unintentionally. After weighing up the risks and benefits, we therefore release them in the interest of reproducibility and of people who wish to build on our work.

The dataset we use, which ultimately derives from the templates collected by Kiritchenko and Mohammad (2018), does not contain any information that names or uniquely identifies individual people or offensive content. Our use of this dataset is consistent with its intended use, to measure gender and racial bias in sentiment analysis systems.

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

## A Model Implementation Details

Monolingual transformer models have 110 million parameters ($\pm$ 1 million) and vocabularies of 30-32k with 768D embeddings. Multilingual models have 179 million parameters, a vocabulary of 120k, with 768D embeddings. We train the monolingual models with the same training settings as preferred in Keung et al. (2020), and allow the pre-trained weights to fine-tune along with the newly initialised classification layer. The multilingual models are trained identically, save that they have a 100x larger learning rate, and learning rate annealing.

All models were trained for 5 seeds, models trained on monolingual data (mono-T, mono-XLT, and distil-mono-T) were checkpointed 15 times. ZS-XLT models were checkpointed 6 times. In total we train 1525 models: 3 monolingual (non-baseline) model types with 5 seeds across 5 languages and 15 checkpoints (1,225 models) and 2 multilingual model types (ZS-XLT, distil-XLT) with 5 seeds and 5 languages and 6 checkpoints (300) models.

This study was done on only the converged models, but all models are released for further study.

**Computational Resources.** Each model was trained on 4 NVIDIA Tesla V100 GPUs with 16GB memory. mono-T and mono-XLT models took 6-8 hours to converge, ZS-XLT and distil-ZS-XLT took 15 hours. This is a total of 620 total hours, or 2,480 GPU hours on our resource.

## B Model Performance

| | Standard | | | Distilled | |
|---|---|---|---|---|---|
| | F1 | Steps | Reference | F1 | Steps |
| ja | **0.62** | 44370 | 0.57 | 0.61 | 60436 |
| zh | **0.56** | 35190 | 0.55 | 0.53 | 43750 |
| de | **0.63** | 36720 | 0.62 | 0.63 | 52621 |
| es | **0.61** | 41310 | 0.59 | - | - |
| en | **0.65** | 27050 | 0.63 | **0.65** | 44285 |
| ZS-XLT | **0.69** | 75000 | 0.68 | 33336 | |

Table 2: F1 at convergence and steps at convergence for standard size and distilled models. Monolingual model performance is measured on the MARC data, ZS-XLT model performance on the US reviews data. Refereence performance taken from Keung et al. (2020), classification accuracy. They don't train monolingual models, so the reference performance is mBERT classification accuracy.

## C Demographics Included in Benchmark Datasets

Racial Minoritised Groups included in the benchmark dataset of Goldfarb-Tarrant et al. (2023) are as below:

For German, this includes Jewish, Roma, Sinti, Arab and Muslim from the UN report, Sorbs as an officially recognised minority, and Polish, Romanian, Turks, Kazakh, Kurds, Russian, Syria, Iraq, Afghanistan, Vietnamese as official large immigrant groups.

For Japanese, this is Chinese, Korean, Okinawan, and generic "Foreign".

For Spanish there is a list of proper names collected from a sociology study that are immigrant names (Salamanca and Pereira, 2013).

For English this is a replication of Kiritchenko and Mohammad (2018) so it is African American proper names.

## D Full set of contingency tables comparing baseline and monolingual models.

The contingency tables for all languages can be shown in Figure 5. A subset of these are included in the main body of the paper.

It is worth noting that saturations are not normalised across all languages and models; this is not a proxy for aggregate comparative bias, it shows the pattern across sentiment scores. The contingency tables also do not show actual (ground-truth) sentiment scores. We include baseline models (left-column) not used in this work for maximum visual comparability to previous work on these benchmarks.

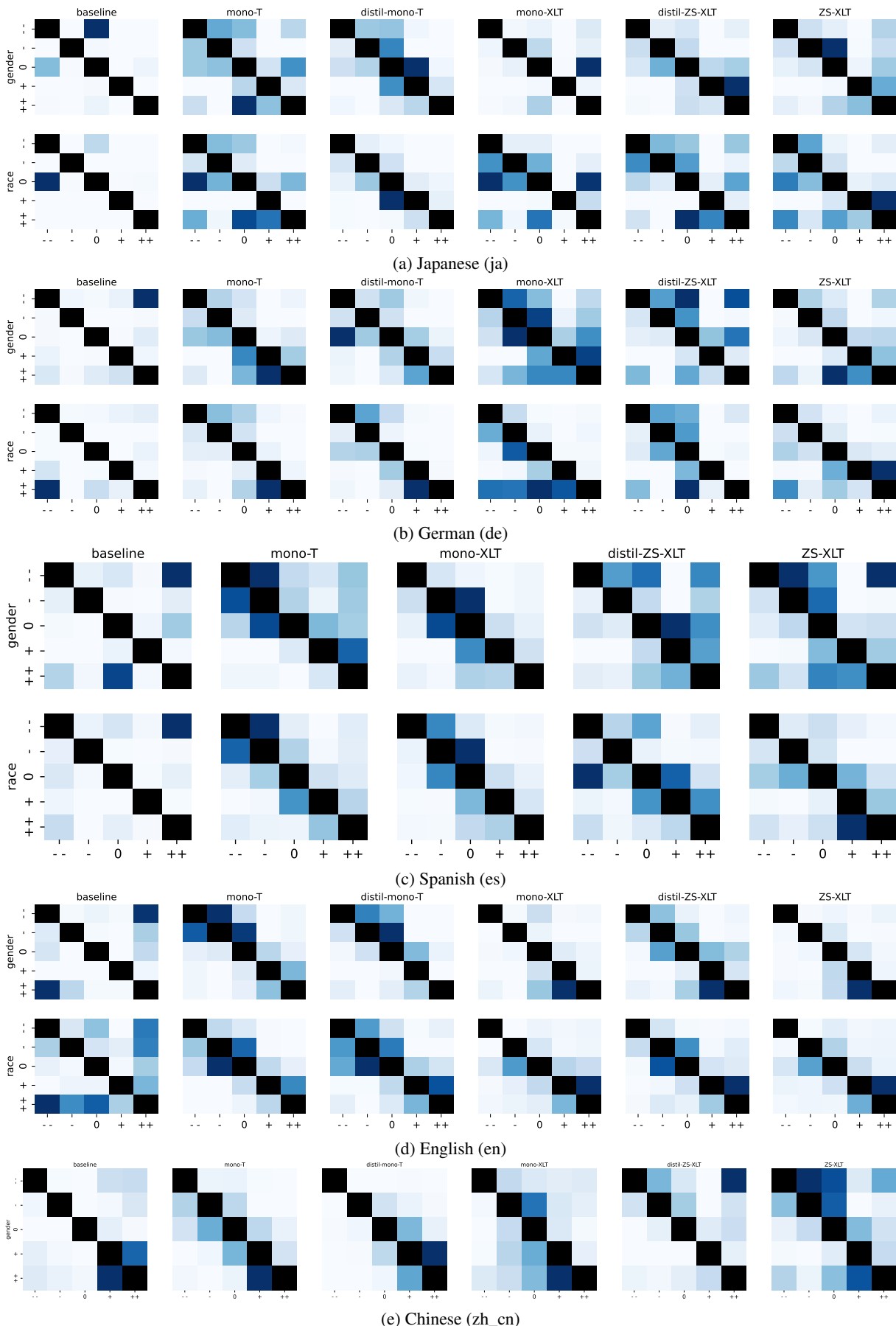

Figure 5: All confusion matrices for experiments in this paper. **++** to **--** are sentiment scores. Rows are predicted sentiment scores for the privileged group, columns predicted scores for the minoritised group. Higher colour saturation in the lower triangle is therefore bias against the minoritised group, in the upper triangle is bias against the privileged group.