# OpenReview forum: "Cross-lingual Transfer Can Worsen Bias in Sentiment Analysis"
_EMNLP/2023/Conference — EMNLP 2023 Main_

### Official Review · Reviewer_hiFd · 2023-08-02

**Typos Grammar Style And Presentation Improvements:** The labels in Figure 3 are extremely …
**Soundness:** 4

**Excitement:**

4: Strong: This paper deepens the understanding of some phenomenon or lowers the barriers to an existing research direction.

**Paper Topic And Main Contributions:**

This paper studies how social biases can transfer across languages during cross-lingual transfer learning for the downstream sentiment analysis task. This analysis is broken down into investigations of bias transfer in the pretraining and finetuning data for five languages. Results show that multilingual pretraining has a larger impact on biases in comparison to pretraining data. In addition, both gender and racial biases are susceptible to transfer.

**Questions For The Authors:**

Why is ZS-XLT finetuned on both MARC and Amazon Customer reviews corpus?

**Reasons To Accept:**

The study of monolingual versus multilingual bias is broken down into pretraining and finetuning data, providing a more comprehensive analysis. In addition, the topic of cross-lingual bias transfer is important and timely, given the increased usage of models in non-English languages.

The authors study languages from a variety of language families.

The conclusion provides a nice overview of the study, while also highlighting specific areas of future research.

**Reasons To Reject:**

There are no examples of racial/immigrant bias subjects in other languages. This makes it hard to determine the specific groups being studied in this paper.

I have some issues with the training procedure. The first is that ZS-XLT is finetuned on additional data compared to other models, which can influence results. The second is that you mention in line 340 that pre-training data can stem from a variety of sources. If this indeed varies across models, this domain shift (e.g. Wikipedia data versus CommonCrawl) can introduce additional uncertainties when comparing models that are supposedly controlled on the pretraining data.

The results reported in Section 6 do not seem to be conclusive and consistent when compared to those shown in Figure 4.

**Reproducibility:**

3: Could reproduce the results with some difficulty. The settings of parameters are underspecified or subjectively determined; the training/evaluation data are not widely available.

**Reviewer Confidence:**

4: Quite sure. I tried to check the important points carefully. It's unlikely, though conceivable, that I missed something that should affect my ratings.

---

> ### Author Rebuttal · Authors · 2023-08-29
>
> Thank you for taking the time to review!
>
> As to your questions and reasons to reject:
> 1. All demographics and emotion words were from the resource created by Goldfarb-Tarrant, et al 2023. We will make this clearer in the text, but we will also use the new space to excerpt them and include examples of them in the text, and the entire list in the appendix. We have also included them below for you.
> 2. ZS-XLT is intentionally trained on additional data, because that is why someone would use a cross-lingual model (to leverage cross-lingual fine tuning data). We also ablate this to control for the effect of the model vs. the effect of the data, which is why we also train mono-XLT models (which are cross-lingual models without additional data). In section 4 lines 302-319 we discuss these choices and why its important to do all of them. This is one of our contributions.
> 3. Re line 340 (domain shift of pretraining data): the pretraining data is consistent within one language, so it doesn’t affect comparisons within one language (which is what we do), it only would across languages, which we don’t focus on. We will clarify this in text. It would be better to have it fully consistent we agree, but that is nigh impossible to control for, since even different versions of commoncrawl can have vastly different data, so we set up our experiments to not require this consistency.
> 4. We will use some of the additional space to break figure 4 into two graphs so they’re easier to read, and to discuss the compression results in more detail with more exact numbers to improve the consistency. They are not fully consistent within themselves as far as trends, but we decided to include them anyway since it’s a valuable research question. Distillation improves bias cross-lingually for gender for English and Japanese, but worsens for German, Spanish, and Chinese, whereas for race it always improve it (save with English where it makes no difference). We weren’t able to work out why this is, but we will revise the text and discuss it further. The important difference is the green to red dot for a practitioner who might want to use cross-lingual transfer to decide whether to distil.
>
> Thank you for the presentational improvement also, we have increased the size of the labels in Figure 3!
>
> The Racial Minoritised Groups:
> For German, this includes Jewish, Roma, Sinti, Arab and Muslim from the UN report, Sorbs as an officially recognised minority, and Polish, Romanian, Turks, Kazakh, Kurds, Russian,  Syria, Iraq, Afghanistan, Vietnamese as official large immigrant groups.
> For Japanese, this is Chinese, Korean, Okinawan, and generic “Foreign”.
> For Spanish there is a list of proper names collected from a sociology study that are immigrant names (Gastã Salamanca and Lidia Pereira. 2013)
> For English this is a replication of Kiritchenko and Mohammad so it is African American proper names.

---

### Official Review · Reviewer_S7SY · 2023-08-04

**Soundness:** 4

**Excitement:**

4: Strong: This paper deepens the understanding of some phenomenon or lowers the barriers to an existing research direction.

**Missing References:**

- https://arxiv.org/pdf/2204.03558.pdf also evaluate gender and racial bias in multilingual settings and creates new datasets for this purpose. Should at least mention this work, even if not evaluating on their data.

**Paper Topic And Main Contributions:**

The authors analyze gender and racial biases in sentiment systems for 5 languages In particular, they are focused on bias differences between monolingual models (models trained directly on the target language, including pretraining data) and cross-lingual models (models trained with multilingual pretraining data and possibly also non-target language sentiment data). They evaluate on existing datasets which consist of pairs of sentences that differ in a single biased term (ex: a gendered pronoun, or a racial term). The authors then measure the score difference between the prediction on the sentence with the privileged group term (ex: male for gender) and the prediction on the sentence with the minority term (ex: female for gender). The authors observe that cross-lingual transfer can worsen bias in sentiment models, and they analyze whether the bias comes from the multilingual pretraining data or from English task data for each target language. The also find that distillation can help to mitigate bias in many but not all cases.

Main contribution: Computationally-aided analysis of bias in multilingual NLP.

**Questions For The Authors:**

A: L494-505 -- this is very confusing. Weak and strong gender agreement are not standard terms so it is not completely clear what is meant here. This seems to suggest that English (which has very little gender agreement) has strong gender agreement, which does not make sense. Can you please clarify?

**Reasons To Accept:**

- Thorough analysis of gender and racial bias in cross-lingual transfer for sentiment in 5 languages. The authors control different sources of bias when possible and examine both aggregate and granular measures of bias. These results are interesting and likely important for researchers working in bias or sentiment.

**Reasons To Reject:**

None

**Reproducibility:**

5: Could easily reproduce the results.

**Reviewer Confidence:**

3: Pretty sure, but there's a chance I missed something. Although I have a good feel for this area in general, I did not carefully check the paper's details, e.g., the math, experimental design, or novelty.

**Typos Grammar Style And Presentation Improvements:**

- L181: "classes to using" --> "classes by using"
- L189: unnecessary semi-colon after "metrics"
- L304: "data in English only" refers to sentiment data right? Should specify this directly.
- Figures 2 and 4 are really difficult to follow in black and white. Please use different symbols in addition to colors to differentiate the models.
- Figure 3: the font labeling each contingency table is way too small to be legible in printed form.
- Figure 4: the spanish gender bias graph only has 3 models, one appears to be missing.

---

> ### Author Rebuttal · Authors · 2023-08-29
>
> Thank you for the review! We’ve made all the presentational improvements, and changed the shapes and label sizes on the graphs.
> We also apologise about the missing Spanish monolingual distilled model – there wasn’t an available pretrained one so we had to leave it out of experiments, and somehow this explanation fell off the caption in one of the edits for length. We have added it back.
> Thank you also for the additional reference that we had missed, we have read and added it also.
>
> Regarding your question on weak/strong gender agreement:  we re-read and see why it is confusing, and will revise. We will define the terms to be more explicit about the extent of grammatical gender marking rather than using ‘weak’ and ‘strong.’
> We intended to say that Chinese and Japanese have very little, German and Spanish have more, and English is in the middle of these two sets, such that when added to Chinese and Japanese gender information is increased and when added to German or Spanish it is decreased. We actually based this theory based on the previous cited work that did this in word embeddings but that isn’t very clear (McCurdy and Serbetci, 2017, and Zhao et al 2020) so we will clarify that also.
> If you have additional suggestions on clarity for this section beyond that, please let us know!

---

### Official Review · Reviewer_F94e · 2023-08-05

**Soundness:** 4

**Excitement:**

4: Strong: This paper deepens the understanding of some phenomenon or lowers the barriers to an existing research direction.

**Missing References:**




**Paper Topic And Main Contributions:**

This paper analyzes the performance of sentiment analysis systems in multilingual setups. The authors compare both mono- and cross-lingual transfer scenarios using counterfactual templates. They run their experiments in five languages. They find that cross-lingual transfer can lead to a more biased model, finding strong especially effects for racial biases.


**Questions For The Authors:**

What is the difference between this study and Goldfarb-Tarrant, et al 2023 which leads to contrasting results in terms of which models pre-trained ones or without pre-training (lines 388-394)? Do you have some assumptions regarding this?

**Reasons To Accept:**

The paper is very well-written, with a clear motivation, experiential setup, presentation of the results, and discussion of the results.
This paper combines two relevant applications, sentiment analysis systems, and cross-lingual training, and test both gender and racial biases for a diverse set of languages.
Experimental setup is sound.


**Reasons To Reject:**

Authors do not provide the list of demographic and emotion words used to create the counterfactual pairs. It is unclear if they were retrieved by the authors of the paper or base on a prior work’s resource.
In lines 287-291, authors mention that the performance of their models converges to be equivalent to previous work (Keung et al, 2020). The results of the prior work should be put in Appendix B next to their results to evaluate this claim.
Authors claim that the larger effect on gender in their experiments comes from the fact that some languages have stronger syntactic gender signal than others. While this statement could be true, it has not been tested in a wider setup in this paper and is rather an assumption given such a small number of languages (Japanese differs not only in gender-marking from German but also culturally which can also have an effect here).
This paper would benefit from stronger statistical testing. Results in Fig 2 could be tested on difference in means. The authors claim that the ZS-XLT models have equal or greater bias but this is not easily spotted for all the languages.


**Reproducibility:**

4: Could mostly reproduce the results, but there may be some variation because of sample variance or minor variations in their interpretation of the protocol or method.

**Reviewer Confidence:**

4: Quite sure. I tried to check the important points carefully. It's unlikely, though conceivable, that I missed something that should affect my ratings.

**Typos Grammar Style And Presentation Improvements:**

The abbreviation SA is introduced already in the abstract, no need to reintroduce it at the beginning of the introduction
The transition between the first and second sentences of the abstract is a bit rough (no connection between sentiment analysis and multilinguality).
Caption Figure 1: The first sentence has a couple of mistakes
4th finding could be made sharper. If compressing models via distillation does not always reduce the bias, you could explicitly give an example when that is the case
Footnote 1 is not there

---

> ### Author Rebuttal · Authors · 2023-08-29
>
> Thank you for the very helpful review!
>
> To address the five listed improvements/reasons to reject:
> 1. The demographic and emotion words were from the resource created by Goldfarb-Tarrant, et al 2023. We will remedy this and make this clearer in the text, but we will also use the new space to excerpt them and include examples of them in the text, and the entire list in the appendix.
> 2. We will include the results of Keung et al 2020 in Appendix B, apologies for the oversight. These are their results, with ours in parentheses for comparison: ja 0.57 (0.62), zh 0.55 (0.56), de 0.62 (0.63), es 0.59 (0.61) en 0.63 (0.65). We exceed slightly in all cases but are very close.
> 3. On reviewing the Case of Gender section, we realise some of what we wrote is unclear. We consider this to be primarily interesting because it is in line with previous work, our citation from McCurdy and Serbetci, 2017 actually has the same finding on word embeddings, that grammatical gender has a larger effect on bias than content, so we find it interesting that our results agree. This is also in line with Zhao et al 2020 in contextual embeddings. Their findings both show that the effect of grammatical gender is stronger than content, which implies that we’d see a change despite the change in culture. However, we agree with you that it is impossible to control for things like culture without creating synthetic data so we will revise this section to clarify that this is a hypothesis that comes from previous work, and that our work strengthens their findings.
> 4. The caption is actually incorrect on Figure 2 from a previous version, our apologies, it still reads “mean and variance” but we had updated it so that the error bars were 95% confidence intervals, so what you see in this review version is in fact confidence intervals. We hope that this satisfies your question. An additional note is that the test sets we use from Goldfarb-Tarrant, et al 2023 are also quite large (3-5k per test) should ensure sufficient power. We will also include the test size stats excerpted from their paper in the final version to make that clear also.
> 5. For figure 2, the comparison of interest for practical bias levels in use is the blue vs. green models (far left to far right) as the orange is essentially an experimental condition allowing us to isolate the contribution of data from that of model. So in comparing those: ja, en, and zh_cn have greater bias for gender, de and es have maybe a bit less but with a larger interval for the ZS-XLT model so we consider them equivalent. For race, only de and es increase, whereas ja and en are equivalent. We will write this out in detail with the additional space. We could also include an additional breakout graph with the orange experiment removed for ease of quick reading.
>
> Regarding the question as to differences between this work and Goldfarb-Tarrant, et al 2023, lines 388-394, their experiments only looked at the effect of pretraining data within a monolingual only setting (no cross lingual transfer). They found that it improved bias to add pretraining data. However, they attributed this finding to the increased stability of the models, since they were comparing to models with no pretraining at all (and thus very little data).
> Our findings on cross-lingual transfer don’t really contradict them, though they do augment and find that the situation is much more complex for cross-lingual transfer and sometimes bias can worsen. We will add this discussion.
>
> We have made all the presentational improvements you suggest, thank you they were very helpful!

---

### Meta-Review · Area_Chair_EffJ · 2023-09-19

**Recommendation:** 4

**Metareview:**

The paper "Cross-lingual Transfer Can Worsen Bias in Sentiment Analysis" presents work on the effect of transfer learning on sentiment analysis in non-English languages and the import of biases from one languages to another.

The arguments against accepting the paper refer to details of the experimental setup.
The arguments for accepting the paper refer to the task as a whole, the depth and extent of the experiments and the results presented.

---

### Decision · Program_Chairs · 2023-10-07

**Decision:**

Accept-Main

**Comment:**

The paper "Cross-lingual Transfer Can Worsen Bias in Sentiment Analysis" presents work on the effect of transfer learning on sentiment analysis in non-English languages and the import of biases from one languages to another.

The arguments against accepting the paper refer to details of the experimental setup.
The arguments for accepting the paper refer to the task as a whole, the depth and extent of the experiments and the results presented.